# Phospholipids in Salt Stress Response

**DOI:** 10.3390/plants10102204

**Published:** 2021-10-17

**Authors:** Xiuli Han, Yongqing Yang

**Affiliations:** 1School of Life Sciences and Medicine, Shandong University of Technology, Zibo 255049, China; hanhxl@sdut.edu.cn; 2State Key Laboratory of Plant Physiology and Biochemistry, College of Biological Sciences, China Agricultural University, Beijing 100193, China

**Keywords:** phospholipids, salt stress, phosphatidic acid, phosphoinositide, phosphatidylserine, phosphatidylcholine, phosphatidylethanolamine, phosphatidylglycerol

## Abstract

High salinity threatens crop production by harming plants and interfering with their development. Plant cells respond to salt stress in various ways, all of which involve multiple components such as proteins, peptides, lipids, sugars, and phytohormones. Phospholipids, important components of bio-membranes, are small amphoteric molecular compounds. These have attracted significant attention in recent years due to the regulatory effect they have on cellular activity. Over the past few decades, genetic and biochemical analyses have partly revealed that phospholipids regulate salt stress response by participating in salt stress signal transduction. In this review, we summarize the generation and metabolism of phospholipid phosphatidic acid (PA), phosphoinositides (PIs), phosphatidylserine (PS), phosphatidylcholine (PC), phosphatidylethanolamine (PE) and phosphatidylglycerol (PG), as well as the regulatory role each phospholipid plays in the salt stress response. We also discuss the possible regulatory role based on how they act during other cellular activities.

## 1. Introduction

Salt (sodium chloride) accumulated in irrigated soil is toxic to plant growth and development, and approximately 40% of irrigated land worldwide is affected by increased salt stress [1]. Nonetheless, plants have developed strategies to cope with salt stress by means of signaling pathways, which include salt sensing, salt signal generation, signal transmission, and minimizing salt damage to plants [2,3].

Phospholipids, which make up the bio-membranes needed for environmental separation between organelles and cytoplasm, and for separation between cells and their environment, also act as signaling compounds in salt stress response [2,4,5,6,7]. Important plant phospholipids include PC, PE, PIs, PS, PA, PG, cardiolipin, and their respective lysophospholipids, such as lysophosphatidic acid (LPA) and lysophosphatidylcholine (LPC). Common features of phospholipids are their non-polar fatty acyl chains, an important component of bio-membranes, and polar head groups, which interact with various factors in the water-soluble environment. Phospholipids have been a key focus of research in recent years, due to their important role in regulating cellular activities such as membrane rearrangement, cytoskeletal dynamics, phosphorus deficiency response, cold stress response, and salt stress response [6,8,9,10,11,12].

The function of phospholipids such as PA and PIs in the salt stress response mechanism has been studied extensively, mainly by analyzing the regulatory function of proteins involved in salt stress signal transduction. The salt stress signal in plants involves salt sensing, response and adaption, and requires the synergy of both plasma and organelle membranes. The determination of the function of membrane phospholipids in salt stress will help to decipher a plant’s tolerance to salt stress. In this review, we will summarize the salt stress signals, as well as the biosynthesis and functions of the major phospholipids, PA, PIs, PS, PC, PE, and PG, in the response of plants to salt stress.

## 2. Salt Stress Signals in Plants

High salinity in plants can cause secondary stresses such as ionic stress, osmotic stress, and oxidative stress. At the same time, however, plants sense and cope with salt stress by means of ionic, osmotic, and reactive oxygen species (ROS) stress signaling pathways, phytohormone signaling, and organelle stress responses, which involve various components, such as phytohormones, osmolytes, organelles, and various proteins.

### 2.1. Salt Stress Perception

The plasma membrane, composed of proteins and lipids, is essential for the perception of salt stress. These components are also involved in triggering an immediate temporally and spatially defined increase of free cytosolic calcium ([Ca^2+^]_cyt_) [2,13]. By using a Ca^2+^-imaging-based genetic screen, one osmotic stress sensor was identified as reduced hyperosmolality-induced Ca^2+^ increase 1 (OSCA1), which is localized at the plasma membrane and forms a Ca^2+^-permeable channel sensitive to osmotic stress [14]. In addition to this, an optimized Ca^2+^-imaging-based genetic screen was used to identify another ionic stress sensor, glycosyl inositol phosphorylceramide (GIPC). This is a sphingolipid localized at the outer layer of the plasma membrane lipid bilayer, which binds Na^+^ outside the cell and mediates the increase of [Ca^2+^]_cyt_ inside the cell [15]. AtANNEXIN4 (AtANN4), a putative Ca^2+^-permeable transporter, is also involved in the increase of [Ca^2+^]_cyt_ under salt stress. However, the activity of AtANN4 is further repressed by the salt overly sensitive (SOS) pathway to fine-tune the [Ca^2+^]_cyt_ signal [16].

### 2.2. Ionic Signaling Pathway

The SOS pathway is a core signal transduction pathway for the salt tolerance of a plant. It is evolutionarily conserved and plays a key role in decoding the salt-induced [Ca^2+^]_cyt_ signal in extruding excess Na^+^ out of the cells. After perceiving [Ca^2+^]_cyt_ by the Ca^2+^-binding proteins SOS3 and SOS3-like Ca^2+^-binding protein 8(SCaBP8), SOS2 kinase activity is activated, and the phosphorylated Na^+^-H^+^ antiporter SOS1 at the plasma membrane can extrude excess Na^+^ out of the cell [17,18]. In this process, the electrochemical gradient established by the plasma membrane H^+^-ATPase (PM H^+^-ATPase) drives SOS1 to transport sodium across the membrane [17,18]. In addition to SOS3/SCaBP8 decoding the salt-induced [Ca^2+^]_cyt_ signal, 14-3-3 proteins also bind and decode the [Ca^2+^]_cyt_ signal to release SOS2 activity [19].

The components of the SOS pathway are regulated to cope with salt stress. When plants grow without salt stress, SOS2 kinase is inhibited by GIGANTEA (GI) and 14-3-3 [20,21], while PM H^+^-ATPase activity is inhibited by SOS2-like protein5(PKS5), SCaBP1, SCaBP3, and PKS24 [22,23,24]. When salt stress acts on a plant, SOS2 activity is released through the degradation of GI and the 14-3-3 repressor, and is also activated by SOS3, SCaBP8, and Geminivirus Rep-Interacting Kinase 1/2 (GRIK1/2) [18,25,26]. SOS1 activity as a result of salt stress can be stimulated by SOS2 and mitogen-activated protein kinase 6 (MPK6) [18,27]. PM H^+^-ATPase activity due to salt stress is regulated by free unsaturated fatty acids and J3 (DnaJ homolog3) [28,29]. As part of this process, free unsaturated fatty acids bind to the C-terminus of PM H^+^-ATPase and stimulate its activity, while J3 inhibits PKS5 kinase activity to release PM H^+^-ATPase [28,29]. When salt stress is eliminated, the SOS pathway will be inhibited to restore plant growth. This can be seen in the negative regulation of SOS2 by brassinosteroid (BR)-insensitive 2 (BIN2) [30].

Besides the SOS pathway, that can exclude Na^+^ from the cytoplasm, an Na^+^/H^+^ exchanger and vacuolar H^+^-ATPase are activated during salt stress to help the Na^+^ compartments inside the vacuoles [31,32]. Maintaining the Na^+^/K^+^ ratio is critical for plant survival during salt stress. Some ion transporters, such as *Arabidopsis* K^+^ transporter 1(AKT1) and high-affinity K^+^ channel (HKT) proteins, contribute to Na^+^/K^+^ homeostasis [33,34,35]. 

### 2.3. Osmotic Signaling Pathway

Abscisic acid (ABA) signaling plays an essential role in the salt stress response of plants [2,36]. The SNF1-related protein kinase 2 (SnRK2) family of proteins are vital to the osmotic stress response [37,38]. It is reported that Snrk2.4 and Snrk2.10 play a positive regulatory role in salt and hyperosmotic stress responses. Their activity is induced by salt, and is localized to cellular membranes through their interaction with PA [39]. Further study revealed that SnRK2.4 and SnRK2.10 regulate root growth during salt stress, while SnRK2.1, SnRK2.5, and SnRK2.9 regulate root growth under non-salt stress conditions through the coordination of auxin signals [40]. The activity of SnRK2.4 can be inhibited by protein phosphatases 2C (PP2C) and ABA insensitive 2 (ABI2), while SnRK2.4 also interacts with v-myb myeloblastosis viral oncogene homolog 21 (MYB21) in the salt stress response of *Arabidopsis* [41,42]. In addition, the SnRK1 family of proteins, which act as an energy sensor in the regulation of cellular metabolism in response to stress, are also related to the salt stress response, since intracellular energy is closely correlated with cellular response to various stresses [25,43,44]. GRIKs are involved in the regulation of phosphorylation and stimulation of SnRK1 in *Arabidopsis* [25].

MPKs, mitogen-activated protein kinase kinases (MKKs), and mitogen-activated protein kinase kinase kinases (MKKKs) are key players in osmotic stress signal transduction [45]. There is growing evidence to show the role they play in salt stress response, such as MPK3, MPK4, MPK6, MKK4, MKK7, MKK9, MKKK20 in *Arabidopsis*, GhMPK17 in cotton, GMK1 in soybean, and MPKs in horticultural plants [27,46,47,48,49,50,51]. MKK1-MPK4 mediates signal transduction in rice, while its level of transcription and activity are induced by salt stress [52]. Abiotic stress-responsive Raf-like kinases (AtARKs), the subgroup B3 of the MAPKKK family, are an essential part of SnRK2-mediated osmotic stress signaling pathways [53]. MAPK cascades participate in ABA-regulated cellular activities, such as stomatal movement and seed germination [54]. The MAPK cascades also play a role in auxin signaling, ethylene signaling, BR signaling, and salicylic acid (SA) signaling, all of which are involved in the salt stress response [36,54].

Furthermore, the accumulation of osmolytes, such as sucrose, proline, and betaine in cytoplasm, also play important roles in the osmotic adjustment of salt stress [2,3].

### 2.4. ROS Signaling Pathways

ROS in plants include hydrogen peroxide (H_2_O_2_), the superoxide anion (O_2_^−^) and the hydroxyl radical (·OH). These are produced very quickly in response to salt stress, and the regulation of ROS homeostasis in plant cells plays a very important role in salt response [3,55,56]. ROS is produced by the plasma membrane respiratory burst oxidase homolog (Rboh)/NADPH Oxidase (NOX) and carries out a dual function in the salt response of plants. First, ROS burst in the early stage of salt stress and act as signal molecules to transmit signals (for example, RbohD and RbohF regulate Na^+^/K^+^ homeostasis under salt stress). Secondly, excess ROS can cause damage to lipids, proteins and DNA, and must be scavenged [57,58].

### 2.5. Phytohormones in Salt Stress Response

The role of phytohormones during salt stress has been thoroughly analyzed in recent studies [2,36]. Both early salt stress signals and later plant growth adaption require stress response hormones, such as ABA, SA, jasmonic acid (JA), ethylene, and growth promotion hormones, such as auxin, cytokinins (CKs), BRs, gibberellin (GA), and strigolactones (SLs) [36]. Besides phytohormones, other small organic molecules such as melatonin, polyamine, GABA, and 5-aminolevulinic acid are all involved in plant salt stress response [59,60,61,62]. 

### 2.6. Organelles in Salt Stress Response

The organelles of the cell wall, endoplasmic reticulum (ER), chloroplast, mitochondrion, and peroxisome are all involved in stress response [18]. The cell wall becomes perturbed during salt stress, affecting the stress resistance of a plant [18]. Salt stress induces the rapid depolymerization of microtubules and the removal of cellulose synthase (CESA) complexes (CSCs) from plasma membranes, which impair the synthesis of cellulose and the cell wall. However, the companion of CESA (CC) proteins in CSCs promotes the reorganization of microtubules and the localization of CESA on the plasma membrane to maintain sustained cellulose synthesis and adapt to salt stress in plants [63]. Salt stress can also lead to ER stress, which is a result of an accumulation of unfolded proteins, and the ER membrane plays a role in stress sensing and signaling [18]. It is recognized that salt stress signals from organelles of the cell wall, ER, chloroplasts, mitochondria, and peroxisome are all integrated in the regulation of cellular activities during salt stress [18].

## 3. PA in Salt Stress

### 3.1. PA Generation in Plants

PA can be produced by the phospholipase D (PLD) pathway, and by the phospholipase C (PLC)-diacylglycerol kinase (DGK) pathway [64]. In the PLD pathway, PA can be produced by the hydrolysis of membrane phospholipids, such as PC and PE [64,65]. In the PLC-DGK pathway, PLC hydrolyzes phospholipids in the membrane to produce diacylglycerol (DAG), which is then converted into PA through a phosphorylation reaction by the catalysis of DGK [66].

### 3.2. PA Functions in Salt Stress

Although PA is a simple, common phospholipid, it has attracted the most attention in terms of plant salt stress signaling. PA in salt stress adaption will be elaborated from the following aspects (Figure 1).

#### 3.2.1. PA Level Increases under Salt Stress

Salt-induced PA accumulation has been confirmed by a large number of studies. Using the ^32^P labeling technique, it was shown that PA and phosphatidylinositol bisphosphate (PIP_2_) levels increased within 30 min of salt treatment [67,68,69]. The ^32^P labeling technique can distinguish PLD-derived PA from PLC-DGK-derived PA. Specifically, PLD-derived PA can be identified by the evaluation of PLD activity. This is performed by the addition of an n-butanol inhibitor for PA generation, but also leads to the generation of a detectable amount of an unnatural product: phosphatidylbutanol (PBut) [64,70]. PLC-DGK-derived PA can be identified by the evaluation of DGK activity, by means of a reaction with ^32^Pi-labled ATP, leading to the generation of detectable radioactive PA [64,70]. Quantitation of the PA level by electrospray ionization-tandem mass spectrometry (ESI-MS/MS) supplies the information of fatty acyl chains in the accumulated PA [27]. A recently developed PA-specific biosensor, PAleon, can monitor the precise spatio-temporal dynamics of PA in living plant cells and tissues during salt stress. This suggests that PA accumulates rapidly within 10 min of salt treatment, mainly in the *Arabidopsis* root. The highest level is found at the tip of the root, the next highest in the maturation zone, with the lowest level found in the differentiation zone [71].

Taken together, it can be seen that PA accumulates with salt stress, indicating the regulatory role of PA in the related signaling.

#### 3.2.2. PLD-Derived PA in Salt Stress Response

The PLD pathway contributes to the rapid accumulation of PA under salt stress. PLD in *Arabidopsis* has identified twelve isoforms: PLDα (1,2,3), PLDβ (1,2), PLDγ (1,2,3), PLDδ, PLDε, and PLDζ (1,2) [72]. It is reported that PLDα1, PLDα3 and PLDδ are involved in the salt stress response, and that their genetic mutations show reduced PA levels, increased Na^+^ levels, and a salt-sensitive phenotype [27,65,67,73].

It has also been reported that PLDα1-derived PA binds and activates MPK6, and phosphorylates the C-terminus of SOS1, thereby squeezing excess Na^+^ from plant cells in *Arabidopsis* [27]. Further studies have shown that PLDα1-derived PA also binds and activates the upstream component of MPK6, MKK7 and MKK9, forming a MKK7/MKK9-MPK6 cascade to phosphorylate SOS1 and extruding excess Na^+^ from plant cells [50]. GMK1, a soybean MAPK, can also be stimulated by PA during the salt stress response [47]. This shows that the MAPK cascade signaling system and SOS pathway can be regulated by PLD-derived PA to improve salt tolerance in plants.

PLDα1-derived PA also binds to the microtubule-associated protein MAP65-1, increasing its activity in enhancing microtubule polymerization and bundling during salt stress. The mutated *pldα*1, which features a decreased PA level, also shows a more disorganized microtubule structure compared with the wildtype [73]. As microtubule depolymerization and reorganization play key roles in cellulose synthesis under salt stress [63], the process may be regulated by PLDα1-derived PA through MAP65-1.

Auxin signaling adjusts root growth to adapt to salt stress, which requires auxin redistribution [36]. PIN-FORMED 2 (PIN2) trafficking and localization regulate auxin redistribution, which can be regulated by PLDζ2 during salt stress in *Arabidopsis* [74]. Further study showed that PLDα1- and PLDδ-derived PA directly bind to PINOID kinase, promoting its accumulation on the plasma membrane and stimulating its kinase activity in phosphorylating PIN2, thereby regulating auxin efflux and auxin redistribution under salt stress in *Arabidopsis* [6]. This shows that the redistribution of auxin under salt stress is closely related to the increase in PA content induced by salt stress.

ABA signaling is central to the salt stress response, and PLD-derived PA regulates ABA signaling through an extensive series of interactions between PA and ABA signaling-related proteins. PLDα1-derived PA binds and inhibits ABI1 activity to promote ABA signaling [75]. It binds NADPH oxidase and regulates ABA-induced ROS generation in *Arabidopsis* guard cells [76]. It interacts with the regulator of the G-protein signaling (RGS1) protein, and inhibits its GTPase-activity accelerating protein (GAP) activity to affect the G-protein regulated ABA signaling pathway [77]. It also interacts directly with sphingosine kinase (SPHK), activating it in producing phytosphingosine-1-phosphate (phyto-S1P). In contrast, SPHK and phyto-S1P act on the upstream of PLDα1 and PA to amplify the ABA signal and mediate ABA responses [78]. PA also binds to the guanine nucleotide exchange factor 8 (GEF8), and is essential for ABA-stimulated GEF8 activity towards ROP7. This indicates PA’s role in ABA signaling through the regulation of GEF8 [79]. Although the link between PLD-derived PA and ABA signaling has been established, whether their interaction functions in salt stress remains unclear. It is reported that Snrk2.4, a member of Snrk2s, can be recruited to the cellular membranes in response to salt stress in *Arabidopsis* through an ABA-independent pathway [39].

Besides *Arabidopsis*, the role of PLD-derived PA in salt stress in other plants has also been studied. Rice suspension-cultured cells treated with salt stress showed increased PLDα activity, as well as increased PLDα protein localization in the tonoplast and plasma membrane [80]. Further genetic studies on PLDα knockdown mutations also indicate that PLDα regulates the rice salt stress response through the mediation of PM H^+^-ATPase activity and transcription level of *OsVHA-A* (which encodes tonoplast H^+^-ATPase), *OSA2* (encoding PM H^+^-ATPase), and *OsNHX1* (encoding TP Na^+^/H^+^ antiporter) [80]. The *phospholipase Dα* gene (*AnPLDα*) from the xerophyte *Ammopiptanthus nanus* is up-regulated by high salt, and its heterologous expression in *Arabidopsis* can improve plant salt tolerance [81]. The heterologous expression of a cucumber phospholipase Dα gene (*CsPLDα*) in tobacco (*Nicotiana tabacum*) enhances its salt tolerance by maintaining Na^+^/K^+^ homeostasis, preventing lipid peroxidation, and accumulating osmoprotective compounds such as proline, soluble sugars, and soluble proteins [82]. In tomatoes (*Lycopersicon esculentum*), *LePLDα1* is transcriptionally up-regulated and the protein is activated upon exposure to salt in cell suspension cultures. However, further genetic analysis showed that the reduced expression of *LePLDα1* does not affect salt tolerance [67]. These results indicate a broad role of PLD-derived PA in the salt stress response.

#### 3.2.3. PLC-DGK-Derived PA in Salt Stress Response

PLC-DGK-derived PA is also involved in the salt stress response. PLCs in plants are divided into two groups: PI-PLCs and non-specific PLCs (NPCs). PI–PLCs catalyze the hydrolysis of PIs, such as phosphatidylinositol (PI), PI 3-phosphate (PI3P), PI 4-phosphate (PI4P), and PI 4,5-bisphosphate (PI(4,5)P_2_). These produce DAG and inositol head groups. NPCs catalyze the hydrolysis of common phospholipids, such as PC and PE, to produce DAG and the corresponding head groups [83,84]. The roles of PI-PLCs and NPCs in the salt stress response will be discussed in the later sections on PIs and PC/PE.

DGKs in *Arabidopsis* comprise 7 members (AtDGK1-7), and in rice comprise 8 (OsDGK1-8) [66,85]. The role of DGKs in the salt stress response will be discussed in the later sections on PI-PLC and NPC.

## 4. PI, PIP, and PIP_2_ in Salt Stress

### 4.1. PI, PIP, and PIP_2_ Generation in Plants

PIs in plants primarily include PI, PI3P, PI4P, PI(4,5)P_2_, and PI 3,5-bisphosphate (PI(3,5)P_2_). Their metabolism in plants has been well characterized and discussed [86], so we will only perform a brief review.

The phosphorylation of PI at position D-3 produces PI3P, and at position D-4 produces PI4P. This is achieved through the catalysis of PI 3-kinase (PI3K) and PI 4-kinase (PI4K), respectively. The further phosphorylation of PI4P at position D-5 generates PI(4,5)P_2_ through catalysis of PI4P-5 kinase (PI4P5K), while the further phosphorylation of PI3P at position D-5 generates PI(3,5)P_2_ by means of catalysis of PI3P-5 kinase (PI3P5K) [86,87]. 

PI3K in mammalian cells are made up of three types (class-I, -II, and -III); however, only the type III PI3K/vacuolar protein sorting 34 (VPS34) has been identified in *Arabidopsis* [88,89]. The PI4K in *Arabidopsis* consists of twelve isoforms, of which the type III subfamily (PI4Kα1, α2, β1 and β2) is involved in the synthesis of PI4P, while the type II subfamily (PI4Kγ1–8) does not harbor PI4K activity [86,90,91]. PI4P5K in *Arabidopsis* are divided into two subfamilies: subfamily B consists of isoforms of PI4P5K1-9, while subfamily A consists of isoforms of PI4P5K10-11 [86,90]. The PI3P5K in *Arabidopsis* contains four isoforms, but only one showed catalytic activity in in vitro experiments [86,90].

### 4.2. PI, PIP and PIP_2_ Function in Salt Stress

Due to the metabolic connection between these PIs, we characterize this part into three sections: PI, PI4P and PI(4,5)P_2_ as the first section, PI3P, PI(3,5)P_2_ as the second section, and PI-PLC as the third section. PIs in salt stress adaption will be elaborated in terms of the following aspects.

#### 4.2.1. PI, PI4P, and PI(4,5)P_2_ in Salt Stress

PI and PI4P are involved in the salt stress response (Figure 2). Our recent study revealed that PI binds the C-terminus of PM H^+^-ATPase and inhibits its activity under normal conditions (without salt stress). However, salt stress does induce the turnover of PI to PI4P, which in turn stimulates SOS1 activity to exclude excess Na^+^ from the cell. This turnover is genetically evidenced by the PI synthase mutations *pis1-1* and *pis1-2*, which feature a reduced PI level and the salt-tolerant phenotype, and also by the PI4P synthase mutation *pi4kβ1*, which has a reduced PI4P level and the salt-sensitive phenotype [92]. Our study reveals that the metabolic turnover of PIs plays a key role in salt stress response.

PI4P and PI(4,5)P_2_ are mostly localized at the plasma membrane in *Arabidopsis*, while PI4P is also found accumulated in the apoplast of tomatoes [93,94]. The large accumulation of PI4P at the plasma membrane is a critical hallmark, since it generates an electrostatic field and controls signal transduction for various cellular activities such as cell development, reproduction, and nutrition [95]. This PI4P-driven electrostatic field may control protein localization on the plasma membrane during salt stress. This includes the auxin signal component PINOID, the BR signal component BRI1 KINASE INHIBITOR1 (BKI1), and the receptor-like kinase signal component membrane associated kinase regulators (MAKRs), which may also be involved in salt stress tolerance [6,95,96]. It is reported that PI4K is also involved in the internalization of clathrin-mediated CESA3 from the plasma membrane, as well as in the internalization and active removal of a membrane-bound cellulase KORRIGAN1 from the plasma membrane [97,98]. The evidence that PI4K participates in membrane trafficking is also supported by its role in lateral root formation through endocytic trafficking [99]. Although the link between the PI4P-driven electrostatic field and membrane trafficking activities during the salt stress response remains unclear, it is possible that PI4P may participate in the salt stress response through the determination of the electrostatic field of the membrane, further influencing its trafficking functions.

It has been reported that SA is also involved in the salt stress response [100,101,102]. PI4K has been found in SA signaling, as a part of which PI4K activity is stimulated by SA, while SA accumulates in the *pi4kIIIβ1β2* mutation in *Arabidopsis* [103,104,105]. PI4P can bind to PLANT U-BOX13 (PUB13) and suppress SA-mediated plant defense signals in uninfected *Arabidopsis* [106]. Whether the PI4P-regulated SA signaling functions as a salt stress response needs further study.

Besides PI4P, its product PI(4,5)P_2_ also participates in salt stress response (Figure 3), and PI(4,5)P_2_ is also localized at the plasma membrane in *Arabidopsis* [94]. The rapid accumulation of PI(4,5)P_2_ was found in salt-stressed *Arabidopsis*, accompanied by an increased level of IP_3_ and [Ca^2+^]_cyt_, indicating that PI(4,5)P_2_ is involved in the salt stress response through its hydrolytic products IP_3_ and [Ca^2+^]_cyt_ [68,107].

Auxin plays a critical role in the response to salt stress, and plants respond by inhibiting root growth and rewiring their root system architecture by means of auxin signaling [2,108]. Auxin treatment modulates PI levels at the plasma membrane with increased PI(4,5)P_2_ and decreased PI4P through the modulation of PI4P5K1 and PI4P5K2 [109]. The clathrin-mediated membrane trafficking regulated by PI(4,5)P_2_ is necessary for polar positioning of PIN proteins, further affecting auxin signaling in cell functions such as root gravitropism [109,110,111]. Since PA also plays a role in the regulation of PIN2-mediated auxin signaling through the regulation of PINOID during salt stress [6], it is unclear whether PI(4,5)P_2_ can coordinate with PA to regulate PIN2 under salt stress conditions. An elucidation of the link between salt-induced PI(4,5)P_2_ accumulation and the auxin signaling can help to further decipher the mechanism of the salt stress response.

The role of PI(4,5)P_2_ in vesicular trafficking during salt stress has been previously reported [112,113] Recent research on *Arabidopsis* shows that the accumulation of PI(4,5)P_2_ in plasma membranes induced by salt stress can act as a molecular signal for the salt stress-induced ER and plasma membrane connectivity [114]. This is achieved by means of PI(4,5)P_2_-mediated synaptotagmin 1 (SYT1)-enriched ER–PM contact site expansion [114]. PI(4,5)P_2_ mediated vesicular trafficking and non-vesicular communication at inter-organelle membrane contact sites under salt stress further support the signal transduction role of phospholipids, in addition to them being a basic component of the membrane structure.

Polyamines play an important role in salt stress tolerance in various plant species such as sugar beet, *Luffa acutangula*, *canola* and *Arabidopsis* [59,115,116,117,118]. Spermine treatment triggers an increase of PI(4,5)P_2_ in *Arabidopsis* root cells, and is a substance involved in regulating K^+^ homeostasis in plants [119]. The plasma membrane-localized polyamine transporter 3 (PUT3) physically interacts with SOS1 and SOS2, while the transport activity of each is regulated under stress conditions in *Arabidopsis* [59]. Whether polyamine signaling is regulated by PI(4,5)P_2_, remains unclear and requires further study.

#### 4.2.2. PI3P, PI(3,5)P_2_ in Salt Stress

PI3P and PI(3,5)P_2_ participate in the salt stress response by regulating various cellular activities (Figure 4). Unlike PI4P, which has the highest plasma membrane content, PI3P is more evenly distributed throughout the endomembrane [94].

PI3K has been reported to play a positive role in the regulation of plasma membrane endocytosis and in the production of intracellular NADPH oxidase-derived ROS during salt stress [120]. The PI3K-regulated ROS generation plays a vital role in regulating rice seed vigor [121], as well as ABA-induced ROS generation and stomata closure [122,123,124]. Although PI3K plays a role in ROS signaling during salt stress, exactly how PI3P regulates ROS production and how PI3K enzyme activity is regulated by salt stress still requires further study.

Autophagy is rapidly induced by salt stress and is required for the salt stress response [125,126,127,128]. PI3P can bind to the BAR-domain Protein SH3P2 and regulates autophagosome formation in *Arabidopsis* [129]. PI3K complex promotes autophagy and the autophagy-related 14 (ATG14)/VPS38 mutation collection permits the study of PI3P function in plant autophagy [89]. PI3P may participate in the salt stress response by regulating the autophagy process, though further research and evidence are needed.

In addition, PI3K is involved in regulating phytohormone-mediated cellular activities. Examples include the promotion of ethylene biosynthesis to accelerate flower senescence in transgenic *Nicotiana tabacum* [130], the regulation of GA-stimulated α-amylase secretion in barley (*Hordeum vulgare*) [131], and the delay of methyl JA (MeJA)-induced leaf senescence in *Arabidopsis* [132,133]. Ethylene, GA, and MeJA are all reported to play regulatory roles during salt stress [134,135,136,137]. Although the link between PI3P and these phytohormone-mediated salt stress responses remains unclear, phytohormones may participate in salt stress partly through PI3P. What is more, PI3K also plays a regulatory role in the proline catabolism of *Arabidopsis*, which may play a role in salt stress response [138].

PI3P5K, which catalyzes the synthesis of PI(3,5)P_2_ from PI3P, mediates the maturation process of the late endosomes and the recycling of auxin transporters, which in turn affects the interaction of endosome-cortical microtubules in terms of their organization and auxin signaling [139,140]. In addition, the recycling of CESA complex from late endosomes is also regulated by PI3P in *Arabidopsis* under salt stress [97]. Many cellular activities in the salt stress response, such as cell wall synthesis, involve this recycling activity, and PI3P and PI(3,5)P_2_ may be the major phospholipids regulating this process..

PI(3,5)P_2_ and PI3P can also directly bind to immunophilin ROF1, an isoform of the FK506-Binding Protein (FKBP) subfamily, to mediate the regulatory role of ROF1 on seed germination under osmotic/salt stress in *Arabidopsis* [141].

#### 4.2.3. PI-PLC in Salt Stress Response

The catalysis of PI-PLC consumes PIs and generates a product of DAG and inositol head groups. PI-PLC in *Arabidopsis* consists of nine constituent parts: AtPI-PLC1 to AtPI-PLC9 [83,84]. The Ca^2+^ signal may mediate the function of PI-PLCs as part of the salt stress response. AtPI-PLC4 in *Arabidopsis* negatively regulates the salt stress response with a salt-hypersensitive phenotype and a more strongly salt-induced [Ca^2+^]_cyt_ signal in its overexpression lines compared with the control [7]. PI-PLC in rice is made up of four components: OsPI-PLC1 to OsPI-PLC4. OsPI-PLC4 positively regulates the salt stress response, with its mutation having a salt allergy phenotype. Its overexpression line has a salt-tolerant phenotype, while the PA and Ca^2+^ signals mediate the OsPI-PLC4-regulated salt stress response [142,143]. OsPLC1 in rice also plays a positive role in the salt stress response. Part of this involves OsPLC1 being recruited to the plasma membrane, hydrolyzing the substrates of PI(4,5)P_2_ and PI4P to generate DAG and IP_3_, and further inducing an increase in the [Ca^2+^]_cyt_ signal [144].

The [Ca^2+^]_cyt_ signal in animal cells is induced by IP_3_, while in plant cells, it is reported that IP_6_ also mobilizes intracellular Ca^2+^, thereby inducing the [Ca^2+^]_cyt_ signal [7,145,146]. The accumulation of PI(4,5)P_2_ and PI(3,5)P_2_ during salt stress, and further [Ca^2+^]_cyt_ signal production can be a strategy for plant salt tolerance. However, the [Ca^2+^]_cyt_ signal is induced more quickly than the accumulation of PI(4,5)P_2_ and PI(3,5)P_2_; the explanation for this is still unknown.

ABA plays an important role in the salt stress response [36]. ABA can induce the PLC1 activity and IP_3_ level in *Arabidopsis*, while salt stress induces the expression of AtPI-PLC1 in *Arabidopsis*, indicating a role of AtPLC1 in the ABA-mediated salt stress response [146,147]. The ABA-regulated stomatal aperture is involved in coping with NaCl-induced osmotic stress for plants, and PI-PLCs (such as AtPI-PLC7 and AtPI-PLC3) have been reported as playing a role in ABA signaling by controlling the stomatal aperture in *Arabidopsis* [36,148,149,150,151,152]. The influence of PI-PLC on the stomatal aperture is also found in the sustained overexpression of *BnPI-PLC2* in transgenic *Brassica napus* seedlings, which showed a reduced stomatal opening [153]. Although PLC activity can be regulated by ABA, the link between the ABA-mediated salt stress response and the PI-PLC-mediated type needs further study.

Plants utilize auxin signaling to reshape their root system architecture, making them better able to cope with salt stress. PI-PLC is also involved in auxin signal transduction [2,108]. AtPI-PLC2 in *Arabidopsis* affects the polar distribution of PIN2 and regulates root development through auxin signaling [154]. Other PI-PLCs, such as AtPI-PLC3 and AtPI-PLC5, have also been reported to be involved in root growth and development [152,155]. Whether PI-PLC-regulated auxin signal transduction participates in salt tolerance remains unclear and needs further study.

Salt stress induces ER stress, and it has been reported that AtPLC2 participates in the ER stress response in *Arabidopsis*. However, whether AtPLC2 functions in the salt-induced ER stress response has not yet been researched [18,156]. Furthermore, proline builds up as a result of salt treatment, and the pharmacological method using U73122 (an Inhibitor of PLCs) can inhibit the accumulation of proline after salt treatment, but will not inhibit it after mannitol treatment, which only discriminates ionic stress and nonionic stress [157]. The regulatory role of PI-PLC in the ER stress response and proline accumulation in the salt stress response needs further study and evidence.

DAG, the product of PI-PLC and NPC, can be converted into PA by means of the catalysis of DGK. Therefore, the PLC pathway and the PLD pathway may act in combination to cope with salt stress [158,159]. It should be noted that PLC consumes substrates and produces substances that exert either a positive or a negative regulatory function during salt stress. How the regulatory roles of the substrates (such as PI4P, PI(4,5)P_2_) and of the products (such as DAG, PA, and IP_3_) can be coordinated in salt stress remains unclear, however, and further research is needed.

## 5. PS in Salt Stress

### 5.1. PS Generation in Plants

PS biosynthesis pathways differ between plant species. There are two common ways that PS biosynthesis takes place in plants: one uses cytidine diphosphate (CDP)-DAG and serine as substrates to synthesize PS by way of catalyzing CDP-DAG-dependent PS synthase (CD-PSS). The other uses serine and a phospholipid (such as PC and PE) to synthesize PS by means of an exchange reaction between phospholipid head groups and serine catalyzed by a Ca^2+^-dependent base-exchange-type PSS (BE-PSS) [87,160].

### 5.2. PS Function in Salt Stress

PS plays a role in the salt tolerance response of plants. PS in salt stress adaption will be discussed from the following aspects (Figure 5).

In *Salicornia europaea* and *Arabidopsis*, salinity induces the accumulation of PS in the plasma membrane, which regulates salt tolerance by maintaining membrane stability and ion homeostasis [161]. In the salt-tolerant cultivar of sweet potato, salinity induces PS accumulation, which delays salt-induced leaf senescence, thus also reducing salt-induced K^+^ efflux and increasing PM H^+^-ATPase activity to positively regulate salt tolerance [162]. Overexpression of phosphatidylserine synthase *IbPSS1* in sweet potato can maintain cellular Na^+^ homeostasis and improve salt tolerance by activating the plasma membrane Na^+^/H^+^ antiporter activity [163].

It is reported that PS, together with PI4P and PA, determines the electronegative features of plasma membranes in plants, which in turn regulate cellular activities by recruiting cytosolic proteins to the specific region of the membrane [164]. The small G protein Rho of Plants 6 (ROP6) can be recruited to the plasma membrane nano-domains by PS, which further participates in auxin signaling, including ROP6-mediated endocytosis and gravitropism [165]. It has been reported that ROP2 is involved in salt tolerance in *Arabidopsis* [166], though it is still not clear whether the PS-regulated electronegative feature of the plasma membrane and ROP6 nano-organization are involved in the salt stress response.

Together, PS participates in salt stress tolerance by regulating PM H^+^-ATPase activity, PM Na^+^/H^+^ antiporter activity, ion homeostasis, and possibly even the electrostatic field of the plasma membrane.

## 6. PC, PE in Salt Stress

### 6.1. PC, PE Generation in Plants

PC can be generated either by the reaction of CDP- choline (CDP-Cho) with DAG, or by a methylation reaction from phosphoethanolamine (P-EA) to phosphocholine (P-Cho) to further produce PC. PE can be generated either by the reaction of CDP-EA (ethanolamine) with DAG, or by a decarboxylation reaction from PS [87,167].

### 6.2. PC, PE Function in Salt Stress

Due to the metabolic depletion of PC and PE by NPC, we characterized this part into two sections: the first section looks at PC and PE in salt stress, while the second section analyzes NPC in salt stress. PC and PE in salt stress adaption will be discussed from the following aspects (Figure 6).

#### 6.2.1. PC and PE in Salt Stress

Although PC and PE are considered structural lipids, it has been reported that salt induces PC production in *Arabidopsis* and maize [168,169]. Choline kinase is required for PC generation during salt stress [170]. PC can interact with the Acyl-CoA-binding protein 1 (ACBP1), and the heterologous expression of an *ACBP1* gene from saline-alkali-isolated algae (*ChACBP1*) in *Arabidopsis* improves plant salt tolerance (possibly via the mechanism of transporting PC to the plasma membrane), increasing PLDδ activity and further converting PC to PE, PS, PG to stabilize the cell membrane [171]. A recent report showed that levels of PC decreased under saline-alkaline stress in maize roots, which suggests an activated PC turnover and lipid reprogramming under saline-alkaline stress [172].

PC biosynthesis is regulated by SnRK1 through the phosphorylation and activity inhibition of CTP:phosphocholine cytidyl transferase 1 in *Arabidopsis* [173]. In addition to this, the transcriptional level of PC and PE biosynthesis genes are regulated under ER stress, while ER-localized choline/ethanolamine kinase 1 is required for ER stress tolerance in *Arabidopsis* [174,175]. These studies indicate that salt-regulated PC production may have some correlation with the cellular energy status and ER stress during salt stress.

#### 6.2.2. NPC in Salt Stress

PC and PE are catalyzed by NPC and generate DAG and head groups of P-Cho and P-EA [83,84]. The NPC family in *Arabidopsis* contains six isoforms, which are named NPC1-NPC6. NPC4, with its localization in root tips and high level of expression in plasma membranes, can be transcriptionally induced and stimulated by salt [176,177]. The stimulation of NPC4 leads to DAG production during salt stress, and its mutation *npc4,* which features a decreased DAG content, shows a salt-sensitive phenotype and compromises the plant response to ABA, indicating the role of PLC-DGK-derived PA signaling and ABA signaling in NPC4-mediated salt response [176,177]. NPC5 in *Arabidopsis* also participates in the salt stress response, as a part of which NPC5 is transcriptionally induced by salinity, with its knockout mutation *npc5-1* showing few to no lateral roots under mild NaCl stress. Exogenous DAG, but not PA, can restore lateral root formation in *npc5-1* under salt stress [5].

DAG, being a product of NPCs, can convert to PA, and further regulates the plant stress response through PA signaling. P-EA and P-Cho, as the other products of NPCs, can produce choline. Choline positively regulates the salt stress response of plants by regulating lipid metabolism and glycine betaine biosynthesis, and this application of choline not only leads to an increase in PC level, but also leads to an increase in other lipids, such as PE, digalactosyldiacylglycerol (DGDG), and monogalactosyldiacylglycerol (MGDG) [178,179,180]. This indicates that PC turnover is activated under salt stress.

Whether PE has any function during salt stress is still unknown, as is any potential mechanism for how this may happen. PC and PE residing in the membrane are considered to be structure lipids, and compared with PA and PIs, their function in salt stress response has not been analyzed in detail. Whether PC and PE play a role in the salt stress response through changes in membrane properties or as signal molecules that regulate proteins, or as substrates that produce signal molecules under salt stress, remains to be seen.

## 7. PG in Salt Stress

### 7.1. PG Generation in Plants

In the process of PG biosynthesis, PA also acts as a substrate to first synthesize CDP-DAG by the catalysis of phosphatidate cytidyl transferase. CDP-DAG in chloroplasts is then converted to phosphatidylglycerol-phosphate (PGP) by the catalysis of PGP synthase, whereupon PGP is further converted to PG by the catalysis of PGP phosphatase [87,181].

### 7.2. PG Function in Salt Stress

Whether PG plays a role in salt stress is still unknown. PG is considered to be a structure lipid of chloroplasts in the thylakoid membrane, a supplier of phosphate under phosphate-limitation conditions, and a regulator of ROP6 in plants [182,183,184]. Unlike other phospholipids, PG is specific to plant cells. PG has also been found in extracellular vesicles, though only in small amounts [185]. Because of the complexity of lipid transport and turnover, whether this anionic chloroplast-lipid (PG) actually regulates abiotic stress, such as salt stress, still needs to be investigated further.

## 8. The Mechanism of Phospholipids in Salt Stress Response

The amphiphilic property of phospholipids determines their localization at various cell membranes. The plasma membrane, primarily consisting of lipids and proteins, can sense extracellular salt signals and respond quickly while initiating a salt stress signal within the cells. Organelle membranes feature a unique composition of lipids and proteins, which can coordinate salt stress signals to adapt to salt stress. The complexity of phospholipids in their head groups and fatty acyl chains determines their multiple functions during cellular regulation.

### 8.1. Phospholipid Head Groups in Salt Stress

The head groups of phospholipids contain different functional groups, such as the single carboxyl group in PS, and the three methyl groups in PC. These all have different configurations (such as PI(4,5)P_2_ and PI(3,5)P_2_) and have different charges (such as PA, PS, PIs, PG carrying negative charges), while PC and PE are neutral phospholipids under physiological pH conditions.

To cope with salt stress, polar head groups with hydrophilic properties are thought to be exposed to the outside of the membrane. In this way, they can communicate with other molecules; an example of this is the interaction between PA and the salt signaling proteins PINOID mentioned above [6]. Besides PA interacting with proteins, other phospholipid-interacting proteins were also identified, such as the interaction mentioned above between PI and the C-terminus of PM H^+^-ATPase [92], PI4P and SOS1 [92], PI4P and PUB13 [106], PI3P and ROF1 [141]. Based on this point, the regulatory role of phospholipids in salt stress is to a certain extent dependent on their binding proteins, such as the regulatory role of PA in salt stress depending in part on its interacting protein PINOID [6]. Our recent work into an improved protein-lipid overlay assay was developed based on the interaction between proteins and phospholipid head groups [186]. To monitor lipid dynamics in living plant cells and explore their roles in the regulation of cellular activities, the biosensors developed for PA, PI3P, PI4P, PI(4,5)P_2_, PI(3,5)P_2_, and DAG were established by the fusion of the lipid-binding domain with fluorescent proteins [71,94,187,188,189,190,191]. The interaction between phospholipids and specific proteins may help the insertion of protein into the membrane, driving membrane bending and scission, along with further vesical formation [192].

In addition to the specific and direct interaction between phospholipids and proteins, phospholipid head groups also affect the electrostatic field of the membrane. Examples of this include PI4P in plants, which affects membrane trafficking activities, the regulation of CESA3 involved in cell wall biosynthesis, and the regulation of PINOID during auxin efflux [95,97]. It is also reported that the divalent metal cations, Zn^2+^ and Ca^2+^, bind to PS and induce membrane blebbing by inducing a contraction of the lipid area [193].

Taken together, phospholipids certainly play a role in salt stress, possibly through the direct interaction between their head groups and salt stress signal proteins, or through their impact on membrane surface characteristics (such as membrane electrostatic fields), to regulate salt stress signaling.

### 8.2. Phospholipid Fatty Acid Chains in Salt Stress

Non-polar fatty acid chains usually have different lengths and different degrees of unsaturation. The analysis of salt-induced PA by ESI-MS/MS revealed that 0.5 h of salt treatment results in an increase in PA level with the fatty acyl chains of 34:2, 34:3, 34:6, 36:3 and 36:6 [27]. C34:2-PA in *Arabidopsis* refers to the fatty acyl chains of 16:0 and 18:2; C34:3-PA means fatty acyl chains of 16:0 and 18:3; C34:6-PA means fatty acyl chains of 16:3 and 18:3; C36:2-PA means fatty acyl chains of 18:0 and 18:2, or 18:1 and 18:1, or both contained; C36:6-PA means fatty acyl chains of 18:3 and 18:3. The difference between the fatty acyl chains at the sn-1 or sn-2 position depends on their biosynthesis pathway: prokaryotic or eukaryotic.

The identification of fatty acyl chains in phospholipids can be performed by ESI-MS/MS, whose fragment ions can be obtained and fatty acyl chains determined, or by gas chromatography–mass spectrometry (GC-MS), whereby fatty acyl chains are determined from their methyl derivatives. The positional identification of fatty acyl chains at the sn-1 and sn-2 positions can be performed by tandem mass spectrometry and then deduced by means of fragment ions. This can be done, for example, via ultra-performance liquid chromatography-electrospray ionization-quadrupole-time of flight mass spectrometry (UPLC-ESI-Q-TOF MS) in the analysis of DGDG [194], or by ^13^C-NMR, which can also identify the position of fatty acyl chains at sn-1 or sn-2 [195]

The single C-C bond of a fatty acyl chain can rotate freely, while an unsaturated double bond cannot. This will change the local membrane properties and further affect cellular activity. Examples of this in plants include the more trienoic fatty acyl chains at low temperatures and the low trienoic fatty acyl chains at high temperatures [196]. The interaction between PA and dynamin-related protein 1 (Drp1), a mitochondrial division regulator in mice (which coincidentally interacts with the head group of PA and then penetrates into the membrane, interacting with the two saturated acyl chains of PA or adjacent phospholipids) is another example of this [197]. The short length and unsaturated nature of the fatty acyl chains will impair this interaction [197]. Similar interactions can be found in plants, such as ENTH domain-containing proteins, which form an amphipathic α-helix 0 (H_0_) upon recognition and binding of PI(4,5)P_2_. The H_0_ helix then inserts itself into the membrane, further causing it to bend, creating membrane curvature and leading to vesical formation [192]. It is also reported that salt-induced osmotic stress can induce the rapid generation of PI4P and PI(4,5)P_2_ with unsaturated fatty acyl chains, which are then quickly depleted to generate IP_3_ to cope with salt stress [198].

Taken together, the fatty acyl chains in phospholipids participate in salt stress signal transduction, possibly through their interaction proteins, or by the effect of their membrane properties.

## 9. Conclusions and Outlook

Plants respond to salt stress through a complex signal transduction involving various proteins and small molecules. Phospholipids, including PA, PIs, PS, PC, PE, and PG, have attracted more and more attention in recent years, due to the important regulatory roles they play in various cellular activities. In this review, we summarize the latest advances in salt stress signal transduction, including ion, osmosis, and ROS signaling pathways, as well as plant hormones and organelles that respond to salt stress. We also summarize the phospholipid functions in salt stress response based on their expression level, the regulation of their enzyme activity, and their biosynthesized content in response to salt stress, as well as by the salt-response phenotype of their genetic mutations. However, information on the role of phospholipids in response to salt stress is quite limited and elusive. How the turnover of phospholipids participates in salt stress response has not been studied in detail. PS, an important regulator in determining the electronegative feature of the plasma membrane in plant cells, needs to be further studied for its regulatory role in salt stress response and other cell activities. The maintenance of the K^+^/Na^+^ ratio as regulated by phospholipids under salt stress is still not very clear, and further research is needed. Although PA and PLC play important regulatory roles in the response to salt stress, studies on the metabolite DAG in plant cells are few and far between. This metabolite plays an important regulatory role in mammalian cells. Phytohormones are also important components for a better understanding of plant salt stress, but their regulatory roles through phospholipids have not been studied in detail. Therefore, in this review, based on the regulatory role of phospholipids in other cell activities, we inferred the possible regulatory role of phospholipids in salt stress response through the regulated components found in salt stress signal transduction. Taken together, this paper summarizes the reported studies on phospholipids in salt stress signal transduction, and also provides information for further research on the regulatory role of phospholipids in response to salt stress.

## Figures and Tables

**Figure 1 plants-10-02204-f001:**
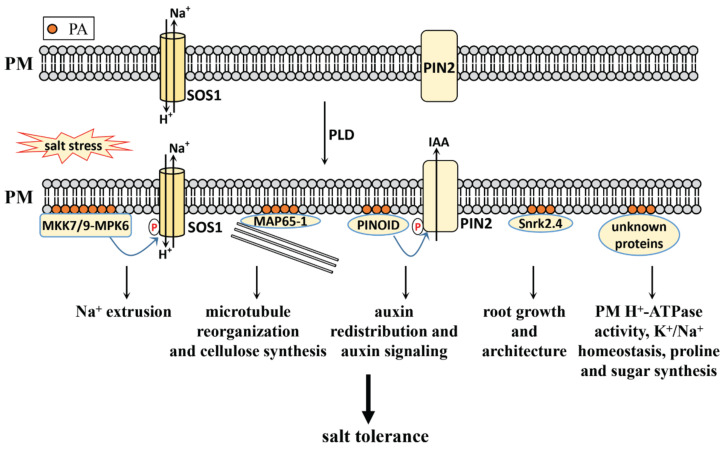
The regulatory role of PLD-derived PA in the salt stress response of plants. PA level is induced by salt stress. PLDα1-derived PA binds and activates MPK6, MKK7, and MKK9 to phosphorylate SOS1 and activate its activity in extruding excess Na^+^ out of the cells under salt stress; PLDα1-derived PA can bind MAP65-1 and regulate it activity in microbutule polymerization and bundling process under salt stress, and this process may regulate cellulose synthesis in salt stress; PLDα-1- and PLDδ-derived PA also binds PINOID and stimulates its activity in phosphorylating PIN2 to participate in salt stress response through the regulation of auxin signaling; PA regulates root growth and architecture through the interaction of Snrk2.4; PA also regulates PM H^+^-ATPase activity, K^+^-Na^+^ homeostasis, proline and sugar synthesis through the unidentified proteins. PM, plasma membrane; PA, phosphatidic acid; SOS1, salt overly sensitive 1; PIN2, PIN-FORMED 2; MPK, mitogen-activated protein kinase; MKK, mitogen-activated protein kinase kinase; MAP65-1, microtubule-associated protein 65-1; SnRK2.4, SNF1-related protein kinase 2.4.

**Figure 2 plants-10-02204-f002:**
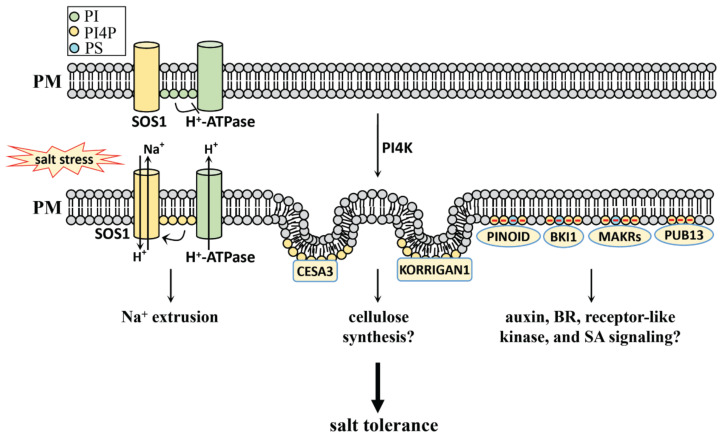
The regulatory role of PI4P in the salt stress response of plants. PI4P level is induced by salt stress. PI binds PM H^+^-ATPase and inhibits its activity under normal conditions without salt stress; however, under salt stress condition, PI4P derived from PI stimulates SOS1 activity and excludes excess Na^+^ from plant cells; PI4P also mediates the internalization of CESA3 and KORRIGAN1 from the plasma membrane, which may participate in salt stress response through the regulation of cellulose synthesis; PI4P at the plasma membrane generates an electrostatic field, which may interact with PINOID, BKI1, MAKRs, and PUB13, and participate in salt stress response. PI, phosphatidylinositol; PI4P, PI 4-phosphate; PS, phosphatidylserine; PM: plasma membrane; SOS1, salt overly sensitive 1; BKI1, BRI1 kinase inhibitor 1; MAKRs, membrane associated kinase regulators; PUB13: PLANT U-BOX13; BR, brassinosteroid; SA, salicylic acid.

**Figure 3 plants-10-02204-f003:**
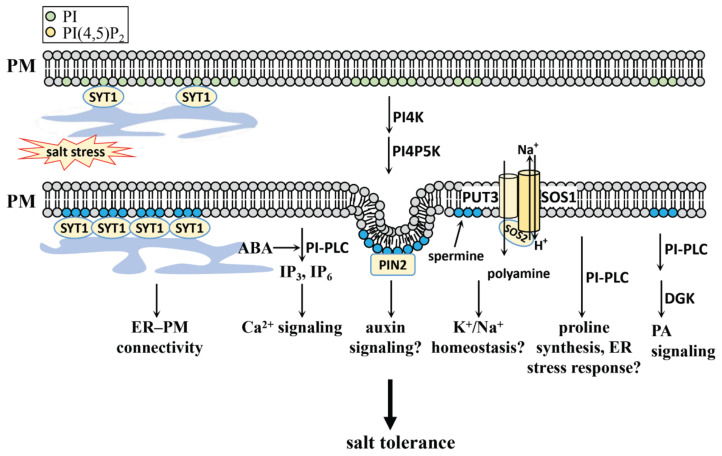
The regulatory role of PI(4,5)P_2_ in the salt stress response of plants. PI(4,5)P_2_ level is induced by salt stress. PI(4,5)P_2_ at the plasma membrane interacts with SYT1 and mediates the salt stress-induced ER and plasma membrane connectivity. The membrane trafficking regulation of PIN2 by PI(4,5)P_2_ may participate in salt stress response through the regulation of auxin signaling. PUT3, a polyamine uptake transporter, interacts with SOS1 and SOS2 under salt stress condition, and the application of spermine can induce PI(4,5)P_2_ accumulation, which indicate a possible link between polyamine, PI(4,5)P_2_ and Na^+^ extrusion under salt stress condition. The hydrolysis of PI(4,5)P_2_ by PI-PLC can produce inositol phosphates, which is involved in Ca^2+^ stimulation under salt stress condition, and ABA may participate in this process by the regulation of PLC activity. PI-PLC may also participate in salt stress response by the regulation of proline synthesis and ER stress response, and the further catalysis by DGK can improve plant salt tolerance through PA signaling. PI, phosphatidylinositol; PI(4,5)P_2_, PI 4,5-bisphosphate; PM, plasma membrane; SYT1, synaptotagmin 1; PM, plasma membrane; ER, endoplasmic reticulum; ABA, abscisic acid; PI-PLC, phosphatidylinositol-phospholipase C; PIN2, PIN-FORMED 2; DGK, diacylglycerol kinase; PUT3: polyamine uptake transporter 3; SOS, salt overly sensitive.

**Figure 4 plants-10-02204-f004:**
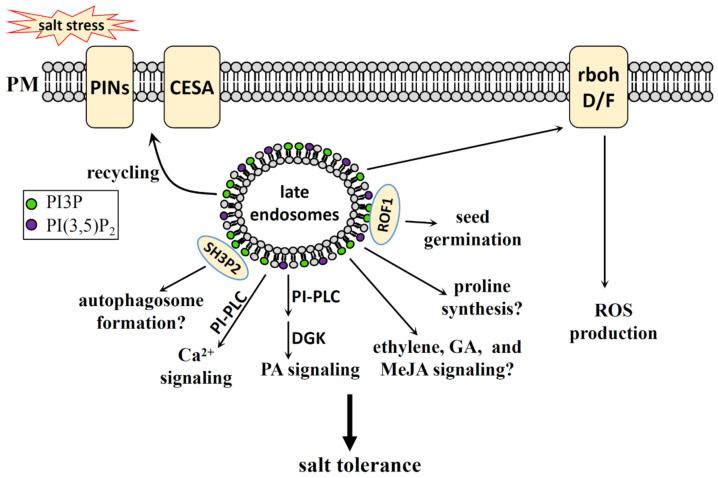
The regulatory role of PI3P and PI(3,5)P_2_ in the salt stress response of plants. PI3P and PI(3,5)P 2 mainly reside in late endosomes. PI3K and its product PI3P play a role in ROS production under salt stress, and rbohD/F may mediate this process. PI3P interacts with SH3P2, which may mediate autophagosome formation under salt stress. PI3K may also participate in salt stress response by its regulation function on ethylene, GA, and MeJA signalings, and proline biosynthesis. PI3P and PI(3,5)P_2_ regulate plant salt tolerance by their regulatory roles on the recycling of PINs and CESA from late endosomes under salt stress. PI(3,5)P_2_ can interact with ROF1 and regulate plant seed germination under salt stress. The hydrolysis of PI(3,5)P_2_ by PI-PLC can produce inositol phosphates, which is involved in Ca^2+^ stimulation under salt stress condition, and ABA may participate in this process by the regulation of PLC activity. The hydrolysis of PI(3,5)P_2_ by PI-PLC and the further catalysis by DGK can improve plant salt tolerance through PA signaling. PI3P, PI 3-phosphate; PI(3,5)P_2_, PI 3,5-bisphosphate; PM, plasma membrane; PIN, PIN-FORMED; CESA: cellulose synthase; SH3P2, SH3 domain-containing protein 2; PI-PLC, phosphatidylinositol-phospholipase C; DGK, diacylglycerol kinase; PA, phosphatidic acid; GA, gibberellin; MeJA, methyl jasmonic acid; ROS, reactive oxygen species; rbohD/F, respiratory burst oxidase homolog D/F.

**Figure 5 plants-10-02204-f005:**
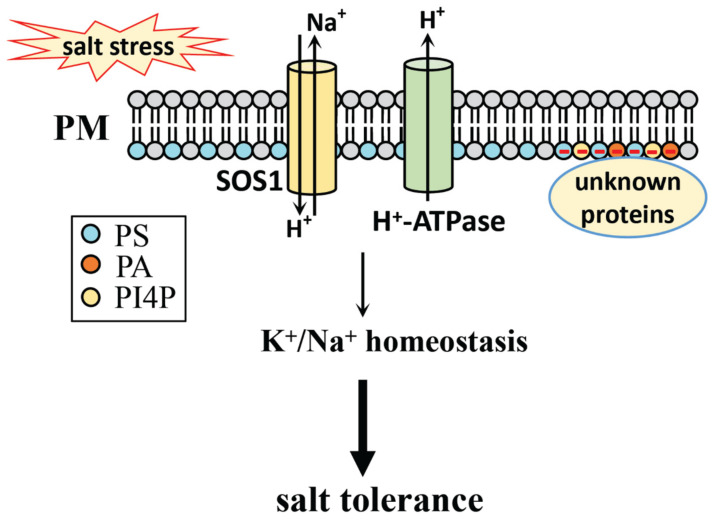
The regulatory role of PS in the salt stress response of plants. PS level is induced by salt stress. PS regulates K^+^/Na^+^ homeostasis by regulating PM H^+^-ATPase activity and SOS1 activity under salt stress. PS, together with PI4P and PA, regualtes the electronegative features of plasma membrane in plants, which may participate in salt stress response through the unknown proteins not identified yet. PS, phosphatidylserine; PA, phosphatidic acid; PI4P, PI 4-phosphate; PM, plasma membrane; SOS, salt overly sensitive.

**Figure 6 plants-10-02204-f006:**
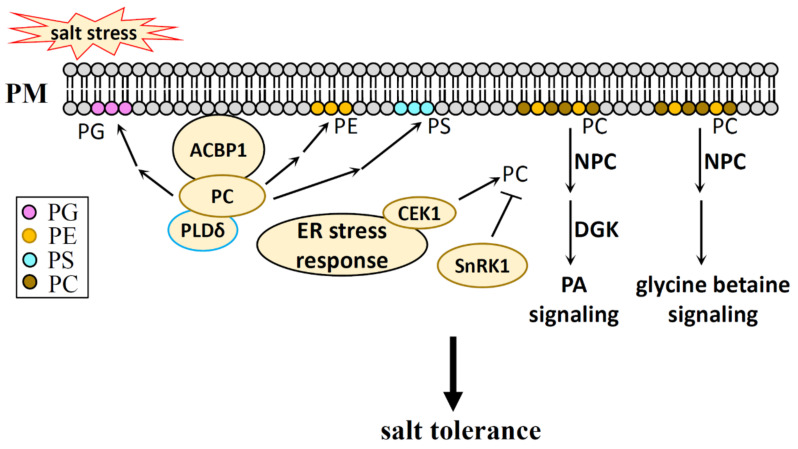
The regulatory role of PC and PE in the salt stress response of plants. PC improves plant salt tolerance by stabilizing the cell membrane through its conversion to PE, PS, PG, and ACBP1 and PLDδ mediate this process; The biosynthesis of PC can be regulated by SnRK1 and ER stress; The further hydrolysis of PC and PE can improce plant salt tolerance through the PA signaling and the glycine betaine signaling. PG, phosphatidylglycerol; PS, phosphatidylserine; PC, phosphatidylcholine; PE, phosphatidylethanolamine; PM, plasma membrane; ACBP1, Acyl-CoA-binding protein 1; PLD, phospholipase D; ER, endoplasmic reticulum; CEK1, choline/ethanolamine kinase 1; SnRK1; NPC, non-specific phospholipase C; DGK, diacylglycerol kinase.

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
