# Peer review of "Phospholipids in Salt Stress Response"

_plants, 2021, doi:10.3390/plants10102204_

Round 1
Reviewer 1 Report
The presented manuscript is an interesting and deep review of the last advances in the study of the role of phospholipids in the response of plants to salt stress. In my opinion, the review is suitable for its publication in the special issue “The Role of Lipid-Hydrolyzing Proteins in Plant Growth”, although, I propose some modifications.
Probably, my most important concern is that I miss a main figure that integrates all the knowledges that the review shows, from the cellular perception of the salt stress to the physiological response, including the different signaling pathways. The figure 1 is too simple.
Most of the time, the complete name of the proteins/genes is given the first time that it is mentioned in the text, and then the correspondent abbreviation is used. However, in a few cases there is only the abbreviation, e.g. ScaBP8 in line 78, MPK6 in line 93, AKT1 and HKT in line 105, PP2C and ABI1 in line 117, etc… The first time you use an abbreviation of a protein/gene/lipid in the text present both the spelled-out version and the short form.
Almost at the end of every paragraph, there is a conclusion to highlight the role of phospholids in the salt stress response, “These results indicate that XXX plays a role in the salt stress response by ….”. These sentences are redundant and can be deleted.
In the section 3.2. PI, PIP and PIP2 function in salt stress, add the correspondent number of the subsections: 3.2.1. in line 322; 3.2.2. in line 391; 3.2.3. in line 433.
Lines 517-519. Add references.
Author Response
Thank you for your constructive suggestions. According to your suggestions, we revised the manuscript.
Review1
(1) The presented manuscript is an interesting and deep review of the last advances in the study of the role of phospholipids in the response of plants to salt stress. In my opinion, the review is suitable for its publication in the special issue “The Role of Lipid-Hydrolyzing Proteins in Plant Growth”, although, I propose some modifications.
Probably, my most important concern is that I miss a main figure that integrates all the knowledges that the review shows, from the cellular perception of the salt stress to the physiological response, including the different signaling pathways. The figure 1 is too simple.
Response: Thank you very much for your constructive suggestion. According to your suggestions, the main figures 2-7 are all revised in the revised manuscript. As to the figure 1, the integrated knowledges on salt stress signaling have been reviewed in our previous publications (Yang and Guo, 2017, Yang and Guo, 2018), and in this review, the figure 1 want to introduce the relationship between plants and their salt stress environment, which is the question we want to study, that how phospholipids respond to salt stress under salt stress environment. Due to the complexity of lipid turnover between different phospholipids, it is not yet clear. Therefore, we reviewed and discussed the regulatory effects of each phospholipid from Figure 2 to Figure 7.
(2) Most of the time, the complete name of the proteins/genes is given the first time that it is mentioned in the text, and then the correspondent abbreviation is used. However, in a few cases there is only the abbreviation, e.g. ScaBP8 in line 78, MPK6 in line 93, AKT1 and HKT in line 105, PP2C and ABI1 in line 117, etc… The first time you use an abbreviation of a protein/gene/lipid in the text present both the spelled-out version and the short form.
Response: Thank you very much for your constructive suggestion. We have given the complete name of the proteins/genes the first time that it is mentioned in the text. Besides the abbreviations you mentioned above, we also checked the manuscript carefully and revised them in the revised manuscript.
(3) Almost at the end of every paragraph, there is a conclusion to highlight the role of phospholids in the salt stress response, “These results indicate that XXX plays a role in the salt stress response by ….”. These sentences are redundant and can be deleted.
Response: Thank you very much for your constructive suggestion. We have revised this problem in the revised manuscript.
(4) In the section 3.2. PI, PIP and PIP2 function in salt stress, add the correspondent number of the subsections: 3.2.1. in line 322; 3.2.2. in line 391; 3.2.3. in line 433.
Lines 517-519. Add references.
Response: Thank you very much for your constructive suggestion. We have revised this problem in the revised manuscript.
Reviewer 2 Report
In the manuscript entitled “Phospholipids in Salt Stress Response”, the authors summarized the generation and metabolism of phospholipid and their roles in salt stress response. A lot of studies showed lipids are close related to abiotic stresses, including salt stress. It is a good perspective to summarize the latest progress of lipids-salt stress crosstalk. However, I found this manuscript is not well organized and summarized. It is more likely a pile of references pasted together than a review paper. I tried hard but found it is not easy to follow and get a clear idea what the real functions of the phospholipids in salt stress. There are many places should be improved.
Major concerns:
(1) I found all the figures in this manuscript are meaningless. All the figures are just some words connect with some arrows. They are too simple to express any meaningfull information to help the reader to understand this paper. I think the figures are one of the key parts of a review. A quantified review should have figures connecting many references together and summarize the key findings to help the reader understand the paper better. I suggest the authors read some classic reviews and learn how others summarizing and making the figures. Obviously, better figures are required for this manuscripts.
(2) I found the structure of this manuscript should be improved. For examples, the authors write several paragraphs to describe the salt stress signals in plants. This paper is focus on “Phospholipids in Salt Stress Response”. It is not necessary to use such a long space to talk the salt signal pathways, which are already well summarized in many reviews.
The authors described the biosynthesis of different phospholipids in each section before the functions of the lipid in salt stress. I found this may make reader confused and can not get a whole picture how these lipids generated. The phospholipids biosynthesis is cross-talked. Some lipids can turn over to each other, and some other synthesis in the same pathways. I suggest the author summarize all the generation of the different phospholipids in one section and make a figure to show how these lipids are synthesized and connected. There are many publications on biosynthesis of phospholipids which the authors can refer to.
(3) For the “Conclusions and Outlook”, I haven’t found any sentence related to “outlook”. The author should summarize the problems of this field and give some prospection.
Author Response
In the manuscript entitled “Phospholipids in Salt Stress Response”, the authors summarized the generation and metabolism of phospholipid and their roles in salt stress response. A lot of studies showed lipids are close related to abiotic stresses, including salt stress. It is a good perspective to summarize the latest progress of lipids-salt stress crosstalk. However, I found this manuscript is not well organized and summarized. It is more likely a pile of references pasted together than a review paper. I tried hard but found it is not easy to follow and get a clear idea what the real functions of the phospholipids in salt stress. There are many places should be improved.
Major concerns:
(1) I found all the figures in this manuscript are meaningless. All the figures are just some words connect with some arrows. They are too simple to express any meaningfull information to help the reader to understand this paper. I think the figures are one of the key parts of a review. A quantified review should have figures connecting many references together and summarize the key findings to help the reader understand the paper better. I suggest the authors read some classic reviews and learn how others summarizing and making the figures. Obviously, better figures are required for this manuscripts.
Response: Thank you very much for your constructive suggestion. According to your suggestions, the main figures 2-7 are all carefully revised in the revised manuscript.
(2) I found the structure of this manuscript should be improved. For examples, the authors write several paragraphs to describe the salt stress signals in plants. This paper is focus on “Phospholipids in Salt Stress Response”. It is not necessary to use such a long space to talk the salt signal pathways, which are already well summarized in many reviews.
Response: Thank you very much for your constructive suggestion. Under salt stress condition, a variety of strategies and proteins have been found in plants to help plants adapt to salt stress. Phospholipids play their regulatory roles mainly through the binding with proteins and the regulation of cellular activities under salt stress. Therefore, as to the part talking the salt signal pathways, although they are already well summarized in many reviews, we reviewed it in brief based on their possible contribution for the regulatory roles of phospholipids in the salt stress response. We also added the latest references in this part. We believe that this part may help readers, including ourselves, understand the regulatory roles of phospholipids, and carry out further study on their regulatory roles in the salt stress response.
The authors described the biosynthesis of different phospholipids in each section before the functions of the lipid in salt stress. I found this may make reader confused and can not get a whole picture how these lipids generated. The phospholipids biosynthesis is cross-talked. Some lipids can turn over to each other, and some other synthesis in the same pathways. I suggest the author summarize all the generation of the different phospholipids in one section and make a figure to show how these lipids are synthesized and connected. There are many publications on biosynthesis of phospholipids which the authors can refer to.
Response: Thank you very much for your constructive suggestion. Because the biosynthesis of phospholipids have been reviewed in some reviews (such as Fatiha, 2019; Heilmann, 2016), and the turn over process between phospholipids is still not very clear. Therefore, in this review, we only support the direct generation pathways of phospholipids, which can be used for researchers to study them by genetic view. We have deleted some parts, which has no direct interaction with salt stress response, such as the de novo biosynthesis of PA. We think the further integration of the biosynthesis of different phospholipids is essential for the deep understanding of their regulatory roles in salt stress response.
(3) For the “Conclusions and Outlook”, I haven’t found any sentence related to “outlook”. The author should summarize the problems of this field and give some prospection.
Response: Thank you very much for your constructive suggestion. We have revised this part in the revised manuscript.
Round 2
Reviewer 2 Report
This manuscript improves a lot compared with the last version. However, I still have some concerns.
The language should be polished. After a brief review of reading I found many mistakes and places are not appropriate and redundant, for examples:
Line 34-38, the two sentence are redundant, and can be expressed using one sentence.
Line 39, “by means of signaling pathways; these include salt sensing”, I think the semicolon(;)is not properly used. It should be “by means of signaling pathways, which include salt sensing”. Moreover, I found the authors use semicolon to connect two separate sentences at many places in the manuscript, for examples, line 78, line 82, line 85, line 192 and so on. I think this punctuation is not properly used in my opinion.
Line 67, “at the same time however,” should be “at the same time, however,”.
Line 71, figure 1 to simple to bring any extra meaning beyond the sentences. I think it can be deleted.
Line 174-178, “Both early salt stress signals and later
plant growth adaption require stress response hormones and growth promotion
hormones; of these, ABA, SA) jasmonic acid (JA), and ethylene are stress response
hormones, while auxin, cytokinins (CKs), BRs) gibberellin (GA), and strigolactones
(SLs) are growth promotion hormones”. These sentences are reduntant. Better use “Both early salt stress signals and later plant growth adaption require stress response hormones, such as ABA, SA, jasmonic acid (JA), ethylene, and growth promotion hormones, such as auxin, cytokinins (CKs), BRs) gibberellin (GA), and strigolactones (SLs).”
……
I really think the authors should carefully read the manuscript and improve the language.
Author Response
Response to reviewer
Thank you for your constructive suggestions. According to your suggestions, we revised the manuscript.
(1) The language should be polished. After a brief review of reading I found many mistakes and places are not appropriate and redundant, for examples:
Line 34-38, the two sentence are redundant, and can be expressed using one sentence.
Line 39, “by means of signaling pathways; these include salt sensing”, I think the semicolon(;)is not properly used. It should be “by means of signaling pathways, which include salt sensing”. Moreover, I found the authors use semicolon to connect two separate sentences at many places in the manuscript, for examples, line 78, line 82, line 85, line 192 and so on. I think this punctuation is not properly used in my opinion.
Line 67, “at the same time however,” should be “at the same time, however,”.
Line 174-178, “Both early salt stress signals and later
plant growth adaption require stress response hormones and growth promotion
hormones; of these, ABA, SA) jasmonic acid (JA), and ethylene are stress response
hormones, while auxin, cytokinins (CKs), BRs) gibberellin (GA), and strigolactones
(SLs) are growth promotion hormones”. These sentences are reduntant. Better use “Both early salt stress signals and later plant growth adaption require stress response hormones, such as ABA, SA, jasmonic acid (JA), ethylene, and growth promotion hormones, such as auxin, cytokinins (CKs), BRs) gibberellin (GA), and strigolactones (SLs).”
……
Response: Thank you very much for your constructive suggestion. According to your suggestions, we found a native speaker to improve the language and polish the manuscript. All the changes are reserved with tracked changes.
(2) Line 71, figure 1 to simple to bring any extra meaning beyond the sentences. I think it can be deleted.
Response: Thank you very much for your constructive suggestion. We have deleted figure 1 in the revised manuscript.
